# The role of RNA in the maintenance of chromatin domains as revealed by antibody-mediated proximity labelling coupled to mass spectrometry

Rupam Choudhury[1], Anuroop Venkateswaran Venkatasubramani[1,2], Jie Hua[1], Marco Borsò[3], Celeste Franconi[4], Sarah Kinkley[4], Ignasi Forné[3], Axel Imhof[1,3]*

[1]Department of Molecular Biology, Biomedical Center Munich, Ludwig-Maximilians University, Planegg-Martinsried, Germany; [2]Graduate School of Quantitative Biosciences (QBM), Ludwig-Maximilians-Universität München, Munich, Germany; [3]Protein Analysis Unit, Biomedical Center (BMC), Faculty of Medicine, Ludwig-Maximilians, University (LMU) Munich, Planegg-Martinsried, Germany; [4]Chromatin Structure and Function group, Department of Computational Molecular Biology, Max Planck Institute for Molecular Genetics, Berlin, Germany

*For correspondence:
lmhof@lmu.de

Competing interest: The authors declare that no competing interests exist.

**Abstract** Eukaryotic chromatin is organized into functional domains, that are characterized by distinct proteomic compositions and specific nuclear positions. In contrast to cellular organelles surrounded by lipid membranes, the composition of distinct chromatin domains is rather ill described and highly dynamic. To gain molecular insight into these domains and explore their composition, we developed an antibody-based proximity biotinylation method targeting the RNA and proteins constituents. The method that we termed antibody-mediated proximity labelling coupled to mass spectrometry (AMPL-MS) does not require the expression of fusion proteins and therefore constitutes a versatile and very sensitive method to characterize the composition of chromatin domains based on specific signature proteins or histone modifications. To demonstrate the utility of our approach we used AMPL-MS to characterize the molecular features of the chromocenter as well as the chromosome territory containing the hyperactive X chromosome in *Drosophila*. This analysis identified a number of known RNA-binding proteins in proximity of the hyperactive X and the centromere, supporting the accuracy of our method. In addition, it enabled us to characterize the role of RNA in the formation of these nuclear bodies. Furthermore, our method identified a new set of RNA molecules associated with the *Drosophila* centromere. Characterization of these novel molecules suggested the formation of R-loops in centromeres, which we validated using a novel probe for R-loops in *Drosophila*. Taken together, AMPL-MS improves the selectivity and specificity of proximity ligation allowing for novel discoveries of weak protein–RNA interactions in biologically diverse domains.

## Editor's evaluation

This important work features an innovative proximity labeling approach to the identification of proteins enriched in distinct types of chromatin domains. The work further shows that selective protein interactions are RNA dependent. This compelling evidence may fundamentally advance our understanding of chromosome domains and the role of RNA in organizing them.

## Introduction

Genetic information in the eukaryotic nucleus is packaged in a complex chromatin structure. Typically, chromatin has been classified in two major categories: euchromatin, which is loosely packed and more accessible, and tightly packed and less accessible heterochromatin (*Heitz, 1932*; *Elgin and Grewal, 2003*). These different degrees of accessibility are thought to be mediated by a network of specific protein–DNA and protein–protein interactions (*van Bemmel et al., 2013*). Systematic mapping studies of proteins on chromatin using DamID (*van Bemmel et al., 2013*; *Filion et al., 2010*) or ChIP (*Roy et al., 2010*) revealed a diverse composition of chromatin resulting in up to 30 different types of chromatin. A variety of different chromatin capture methods have demonstrated that the genome can be separated in two large functional compartments (*Lieberman-Aiden et al., 2009*) with inter-chromosomal interactions occurring mainly between loci belonging to the same compartment. While it is generally accepted that the three-dimensional organization of nuclear chromatin in distinct domains plays a major role in regulating gene expression (*Pombo and Dillon, 2015*; *Reiff et al., 2022*; *Lanctôt et al., 2007*), the mechanisms of how the different types of chromatin form and how they contribute to the dynamic regulation of gene expression are still not fully understood. Several studies have shown that individual chromosomes occupy distinct areas within the nucleus, often called territories (*Cremer and Cremer, 2001*; *Fritz et al., 2019*) and specific domains within chromosomes likewise cluster to the interior or periphery of the nucleus (*Ou et al., 2017*). Prominent examples of such regions are the centromeric heterochromatin that frequently localizes at the nuclear periphery (*Taddei et al., 2004*) or nuclear bodies such as the nucleolus (*Caudron-Herger et al., 2016*) or Cajal bodies that are formed on the rDNA locus (*Caudron-Herger et al., 2016*) or at sites of snRNP and snoRNP biogenesis, respectively (*Cioce and Lamond, 2005*). Nuclear bodies and distinct chromosome territories dissolve at the onset of mitosis and re-form in G1 (*Cheutin et al., 2003*; *Sexton et al., 2012*). Such self-organized formation and maintenance of functional domains are most likely mediated by multiple interactions between the DNA, proteins, and RNA found within these domains (*Quinodoz et al., 2021*). The individual interactions that drive this process are often weak in nature but nevertheless able to drive the formation of distinct nuclear bodies. Though microscopically detectable, many nuclear bodies are highly dynamic and difficult to purify. In fact, even for the most abundant classes of nuclear bodies such as nucleoli or Cajal bodies, the investigation of their proteomic composition required large quantities of cultured cells (*Lam et al., 2002*; *Andersen et al., 2002*; *Andersen et al., 2005*). The purification and characterization of specific chromosomal territories are even more challenging and often involve the disruption of the nuclear 3D structure before purification of a chromosomal domain (*Iglesias et al., 2020*; *Vermeulen and Déjardin, 2020*; *Saksouk et al., 2014*). Therefore, these purification methods frequently depend on the stable interaction of the protein with the DNA and hence many weak interactions are lost. A possible solution to this loss is the use of proximity biotinylation methods (*Roux et al., 2012*), which have been shown to provide powerful tools to identify and characterize such weak interactions (*Youn et al., 2018*; *Samavarchi-Tehrani et al., 2020*). In fact, proximity biotinylation has been increasingly used in chromatin research (*Kochanova et al., 2020*; *Minderjahn et al., 2020*; *Santos-Barriopedro et al., 2021*; *Remnant et al., 2019*; *Villaseñor et al., 2020*) to characterize the chromosomal environment of DNA-bound factors that have been elusive to ChIP or ChIP-MS methods. A major disadvantage thus far of these methods is the requirement of exogenously expressed fusion proteins such as BioID or variants thereof or with Apex2 and new cell lines had to be established (*Kochanova et al., 2020*; *Minderjahn et al., 2020*; *Hung et al., 2014*; *Rhee et al., 2013*). Moreover, as the expression of transgenic fusion proteins and the delivery of biotinylation reagents are difficult to control, it results in a high background and variance of the proximity proteome. To overcome these issues, various in vitro methods using BirA derivatives (*Santos-Barriopedro et al., 2021*; *Remnant et al., 2019*; *Bar et al., 2018*) or secondary antibodies coupled to horseradish peroxidase (HRP) *Bar et al., 2018*; *Hashimoto et al., 2012*; *Kotani et al., 2008*; *Rees et al., 2015* have been established to facilitate proximity biotinylation in cells. The disadvantages of BirA, HRP, and their derivatives are their relatively low biotinylation efficiency and slower kinetics resulting in a lower sensitivity and the requirement of large amounts of input material. In addition, Apex2 allows the efficient labelling of proteins and RNA molecules in proximity to a given bait using biotin-anilin as substrate. To improve the specificity and sensitivity of these approaches we developed a novel method called AMPL-MS (antibody-mediated proximity labelling coupled to mass spectrometry), using a protein A–Apex2 fusion protein. A related approach has recently been

used to study the proteomics composition of protein domains carrying specific histone modifications (*Li et al., 2022*). However, our approach investigates not only proteins but also RNAs in proximity of nuclear factors that define distinct chromosomal domains. Here, we use AMPL-MS in *Drosophila* and show that specific nuclear domains are enriched for specific RNAs, which are most likely transcribed from the DNA that is part of the domain. Furthermore, we show that removal of the RNA substantially changes the proteomic composition and domain morphology (*Li et al., 2022*). Interestingly, deeper evaluation of centromeric RNA–protein proximity data, revealed an important role of R-loops in the formation/maintenance of these domains, which we could confirm using novel orthoganol approaches. Taken together, AMPL-MS is highly versatile and improves the specificity and selectivity of proximity biotinylation, facilitating the discovery of novel and relevant weak RNA–protein interactions within biologically distinct domains.

## Results and discussion
### Versatile proximity biotinylation in cells

To establish a versatile and sensitive method to characterize the molecular neighbourhood of a given chromatin protein, we expressed a His-tagged fusion protein of protein A and Apex2 in *E. coli* (pA-Apex2; *Figure 1—figure supplement 1*). The fusion protein was used to tether Apex2 to chromatin domains marked by specific antibodies, which allows an efficient biotinylation of all proteins in the vicinity (*Figure 1a*). To assess the specificity of the method, we first targeted centromeric chromatin using an antibody against the centromeric histone variant of H3, Centromer Identifier (CID or dCenpA). The colocalization of the biotin signal with CID in immunofluorescence images shows that pA-Apex2 biotinylates proteins within the centromeric domain only in presence of an anti-CID antibody, biotin-phenol, and hydrogen peroxide (*Figure 1b, c*). Hardly any biotinylation is observed when either antibody, biotin-phenol, or hydrogen peroxide is omitted. Due to the presence of endogenous peroxides, we could observe some minor background biotinylation when we did not exogenously add hydrogen peroxide and therefore compared all proximity proteomes to the controls where no primary antibody was added. Proteins within this domain were then isolated using streptavidin beads and analysed by mass spectrometry leading to the identification of 172 proteins that localized in proximity to CID containing centromeric chromatin (*Figure 1d* and *Figure 1—figure supplement 1*). All previously characterized centromeric proteins of *Drosophila* were almost exclusively detected in proximity to CID (*Figure 1e*; *Kochanova et al., 2020*; *Barth et al., 2014*). A gene ontology (GO) enrichment analysis of the proteins that had not yet been reported as CID interactors revealed a strong enrichment of factors involved in RNA-related processes (*Figure 1f* and *Figure 1—figure supplement 2*). A comparison with a proximity proteome experiments performed in SL2 cells expressing a CID-APEX2 fusion protein (*Kochanova et al., 2020*) also showed that the AMPL-MS approach has a much higher sensitivity as we got a comparable number of identified components of centromeric chromatin from as little as $2 \times 10^7$ cells as opposed to $2 \times 10^{10}$ cells in the conventional experiment (*Kochanova et al., 2020*; *Figure 1—figure supplement 2*). This increased sensitivity is most likely due to a much lower background of the AMPL-MS approach thanks to the isolation of nuclei and the extensive washing steps after the incubation with pA-APEX2. These steps greatly reduce interfering background and, thanks to the nuclear permeabilization, increase the penetration of biotin-phenol, which is otherwise not taken up very efficiently by these cells. The establishment of AMPL-MS as a sensitive and versatile method to analyse chromatin domains allows a quick investigation and comparison of such domains even in difficult to isolate cell populations.

### AMPL-MS allows the distinction of different chromatin domains

To demonstrate the applicability of AMPL-MS to characterize different chromosomal territories, we next investigated the proteomic neighbourhood of the transcriptionally hyperactive X chromosome in *Drosophila* (*Figure 2*). To do this, we used an antibody recognizing MSL2, a component of the dosage compensation complex that specifically and selectively associates with the male X chromosome (*Lucchesi and Kuroda, 2015*). Like the selective biotinylation of the centromere, AMPL-MS using an anti-MSL2 antibody resulted in the specific labelling of the X chromosome territory (*Figure 2a, b*) only when antibody, biotin-phenol, and hydrogen peroxide is present. As expected, the composition of the enriched sets of protein depends on the bait identity (*Figure 2c*; *Figure 2—figure supplement 1* and

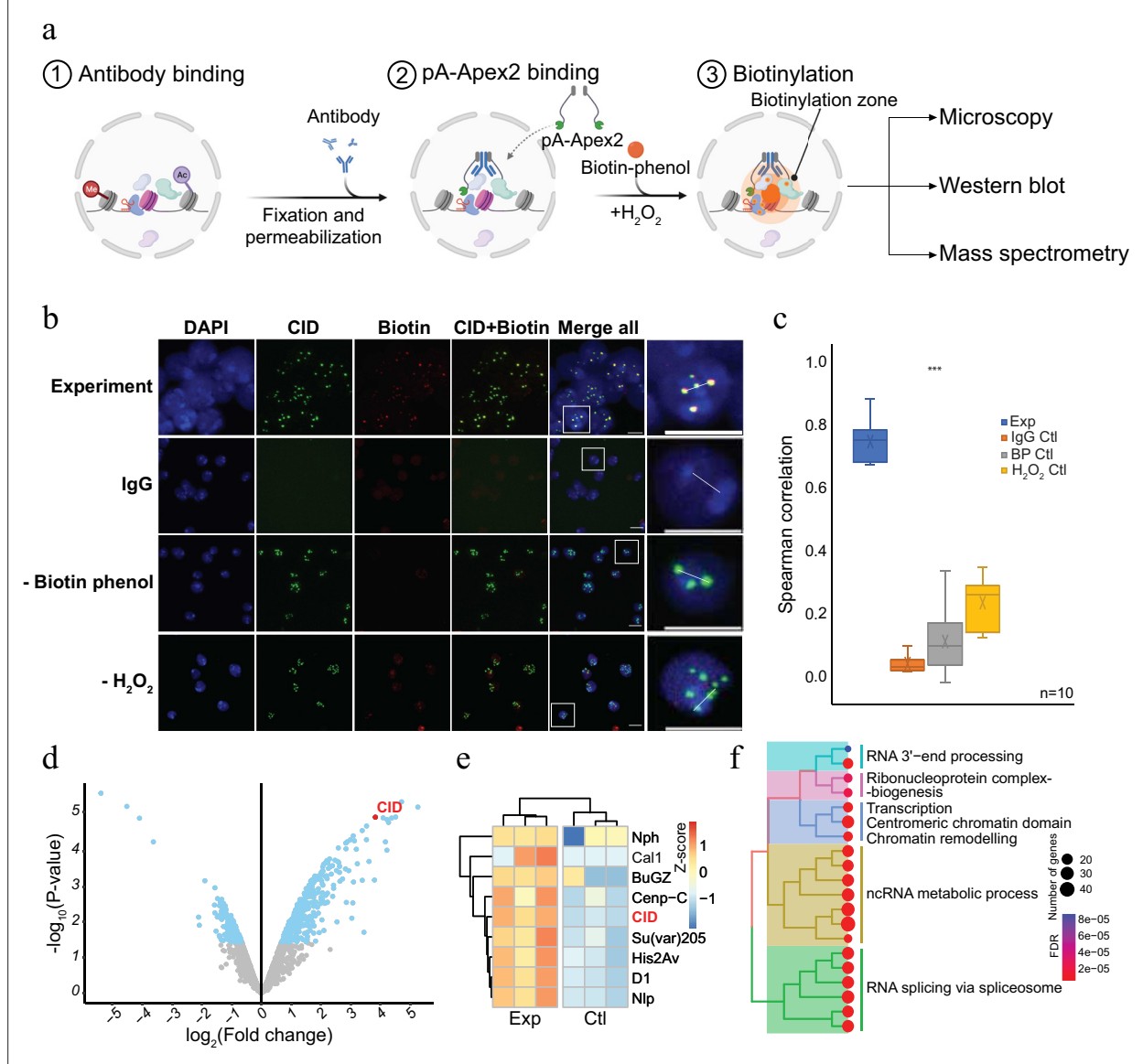

**Figure 1.** Antibody-mediated proximity labelling coupled to mass spectrometry (AMPL-MS) to study the proteomic composition of the *Drosophila* centromere in a genetically unperturbed cell line. (**a**) Schematic of AMPL-MS. Isolated nuclei were fixed, permeabilized, and incubated with specific antibodies. Recombinant pA-Apex2 enzyme binds to the antibody and biotinylates associated proximal proteins upon the addition of $H_2O_2$ and biotin-phenol. Proximal protein biotinylation is visualized by microscopy, western blot and proteins were identified by mass spectrometry. Created with BioRender.com. The schematic (**b**) immunofluorescence microscopy of centromeres using a Cenp-A (CID) antibody (in green) and the corresponding proximity proteome after biotinylation by pA-Apex2 (in red). Nuclear DNA was stained by 4′,6-diamidino-2-phenylindole (DAPI, in blue). IgG was used as antibody control. Scale bars represent 10 µm (large panel) or 5 µm (small panel). (**c**) Distribution of pair-wise Spearman correlations for quantifying the relationship between CID and biotinylation. Images of 10 cells from three independent experiments were used. Statistical significance is based on Wilcoxon rank sum test (***p-value <0.001). (**d**) Volcano plot of purified biotinylated proteins identified by mass spectrometry. The bait protein is highlighted in red. The *x*-axis represents the $\log_2$ fold change and the *y*-axis represents $-\log_{10}$ p-value comparing three IgG replicates with three CID antibody replicates (paired). The significantly enriched proteins (abs(LFC) >1 and $p_{adj} \leq 0.01$) are highlighted in blue. (**e**) Heatmap showing the enrichment of known centromeric proteins. The heatmap was plotted using scaled $\log_2$ raw intensities. Each column represents values obtained from three independent biological replicates. (**f**) Over representation analysis showing top 20 biological processes (BP) using significant proteins from (**d**). Unsupervised clustering was performed for the gene ontology (GO) terms. The colour gradation form blue to red represents FDR (false discovery rate) and dot size represents the number of proteins found enriched in the named pathway (count).

The online version of this article includes the following source data and figure supplement(s) for figure 1:

*Figure 1 continued on next page*

*Figure 1 continued*

**Figure supplement 1.** Establishment of protein A–Apex2- (pA-Apex2) mediated biotinylation for antibody-mediated proximity labelling coupled to mass spectrometry (AMPL-MS) (related to *Figure 1*).

**Figure supplement 1—source data 1.** Raw image files of the *Figure 1—figure supplement 1*.

**Figure supplement 1—source data 2.** Raw image files of the *Figure 1—figure supplement 1*.

**Figure supplement 2.** Antibody-mediated proximity labelling coupled to mass spectrometry (AMPL-MS) for *Drosophila* centromere (related to *Figure 1*).

---

*Supplementary file 3*). Known centromeric proteins were more enriched in the anti-CID AMPL-MS proteome and proteins known to localize the hyperactive X chromosome were mainly detected in the anti-MSL2-AMPL-MS proteome (*Figure 2d*). The analysis of the entire dataset by a principal component analysis also showed a tight clustering of the domain proteome according to the bait identity (*Figure 2—figure supplement 1*). A GO enrichment analysis of the differentially enriched proteins revealed a bias for the GO terms spindle assembly and chromatid segregation in proteins close to CID and sex determination and dosage compensation in the proteins proximal to MSL2 (*Supplementary files 4 and 5*). Interestingly, both chromatin neighbourhoods contained factors related to RNA-related processes albeit with slightly different functional terms (*Figure 2e, f*). Importantly, this method does not allow us to distinguish whether the association of these factors is directly caused by the recruitment of MSL2 to the hyperactive X or whether their localization is caused by the general upregulation of gene activity on the single male X chromosome. As both proteins are tightly associated with chromatin, we reasoned that the histones in proximity to CID should carry different modification patterns than the one in the neighbourhood of MSL2. Indeed, we detected a moderate increase of activating marks in the MSL2 proximity proteome and a slightly higher level of repressive marks in proximity of CID (*Figure 2g*). These comparative experiments show that we can apply AMPL-MS to distinguish distinct chromatin domains based on their proteomic composition.

## Proximity labelling of proteins associated with post-translationally modified histones

Having established AMPL-MS as an efficient and sensitive method to identify proteins associated with distinct chromatin domains marked by antibodies that recognize signature proteins, we next applied AMPL-MS to characterize the proteome in proximity to specific histone marks. To this end, we used antibodies recognizing H3K4me3, H3K9me3, and H4K16Ac for AMPL-MS (*Supplementary files 6–8*). Consistent with the role of these histone modifications in establishing transcriptional active (H3K4me3 and H4K16ac) or repressive (H3K9me3) chromatin domains, the proteomic composition of these domains are very different (*Figure 3a* and *Figure 3—figure supplement 1*). As expected, proteins in proximity to H3K9me3 includes several known K9me3-binding proteins, such as Su(var)205 (HP1a) or HP5 and other known heterochromatin-associated proteins such as Su(var)3–7, Su(var)3–3, HP5, HDAC3, HDAC1, and HDAC6 (*Figure 3b*). Consistently, we also find an overlap with the centromeric H3 variant CID and proteins that we detected in proximity of it like NPH, NLP, or CENP-C. In contrast, we find mainly proteins associated with active chromatin and the components of the dosage compensation complex in proximity of H4K16ac (*Figure 3b*).

While most of these factors have already been described before as bound to repressive or active chromatin in other systems, such a systematic analysis has not been done in the *Drosophila* system. Moreover, the highly sensitive AMPL-MS method also allowed us to investigate a number of novel factors involved in splicing and RNA processing that are selectively detected in the neighbourhood of H3K9me3 or H4K16ac containing chromatin (*Figure 3c*). Despite having the same GO term associated with, the individual proteins in proximity to H3K9me3 or H4K16ac were quite different (*Figure 3d*). Factors like SAF-B or HNRNP-K, which we detect in proximity to H3K9me3 have been shown to play an RNA-dependent role in heterochromatin organization (*Huo et al., 2020*) and in RNA-mediated transcriptional silencing (*Pintacuda et al., 2017*). Several RNA-binding proteins we detect closer to H4K16 on the other hand are part of the canonical or non-canonical splicing machinery (*Ustaoglu et al., 2019*) and are often found close to sites of active transcription. The abundance of specific RNA-binding proteins in proximity to various chromatin domains suggests a major and specific role of RNA in the organization of chromosomal domains.

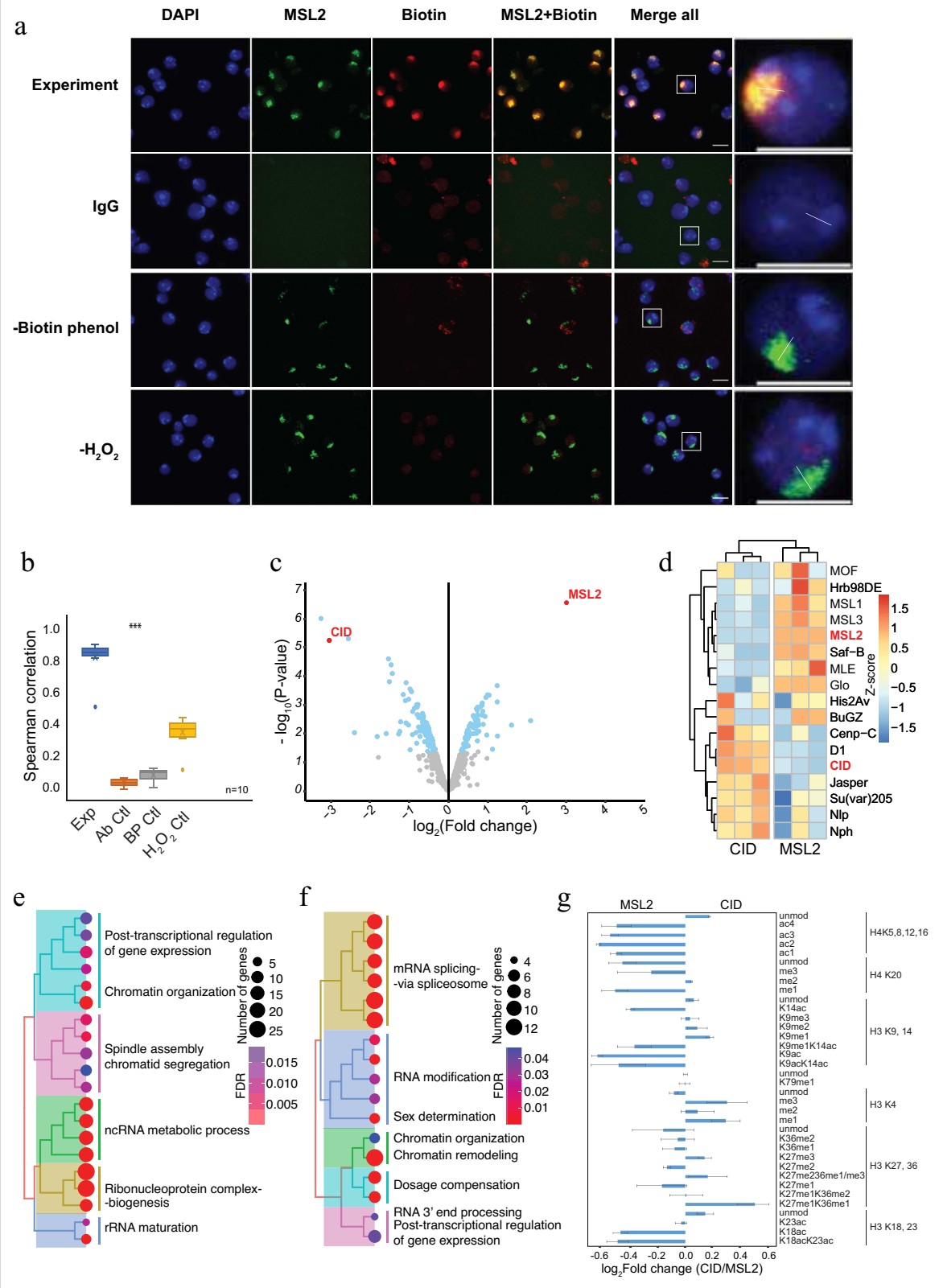

**Figure 2.** Antibody-mediated proximity labelling coupled to mass spectrometry (AMPL-MS) can efficiently identify protein associated with different chromatin domains. (**a**) Immunofluorescence of the X chromosome bound by MSL2 antibody (in green) and biotinylated proteins after biotinylation by pA-Apex2 (in red). Nuclei were stained by DAPI (in blue) and IgG was used as antibody control. Scale bars represent 10 μm (large panel) or 5 μm (small panel). (**b**) Ten cells from three independent experiments were used to quantifying the relationship between MSL-2 and biotinylation. Images of 10 cells

*Figure 2 continued on next page*

*Figure 2 continued*

from three independent experiments were used. Wilcoxon rank sum test is used for comparison (***p-value <0.001). (**c**) Volcano plot of biotinylated proteins identified by mass spectrometry. The bait proteins are highlighted in red. The *x*-axis represents the $\log_2$ fold change and the *y*-axis represents $-\log_{10}$ p-value comparing three CID replicates with three MSL-2 antibody replicates (paired). The significantly enriched proteins ($\log_2$fold change(LFC) >1 and $p_{adj} \leq 0.01$) are highlighted in blue. (**d**) Heatmap displaying the enrichment of known centromeric and X-chromosome-associated proteins. The heatmap was plotted using scaled $\log_2$ raw intensities. Each column represents values obtained from three independent biological replicates. Over representation analysis showing top 20 biological process (BP) for significantly enriched proteins from (**c**) associated with either centromere (**e**) or the X chromosome (**f**). The colour gradation form blue to red represents FDR (false discovery rate) and dot size represents the number of proteins found enriched in the named pathway (count). (**g**) Relative quantification of histone modifications associated with the X chromosome territory (left) and the centromere (right).

The online version of this article includes the following source data and figure supplement(s) for figure 2:

**Figure supplement 1.** Proximity proteomics for *Drosophila* X chromosome and its comparison to the centromeric chromatin domain (related to *Figure 2*).

**Figure supplement 1—source data 1.** Raw image files of the *Figure 2—figure supplement 1*.

**Figure supplement 1—source data 2.** Labelled raw image files of the *Figure 2—figure supplement 1*.

## The removal of RNA changes the proteomic environment of chromatin domains

Chromatin-associated RNA has been suggested to serve as an architectural component or by facilitating the formation of membrane less condensates within the nucleus (*Schubert et al., 2012*; *Żylicz et al., 2019*; *Thakur and Henikoff, 2020*; *Sabari et al., 2020*; *Lyon et al., 2021*; *Guo et al., 2021*). Many of the RNAs can act in cis as well as in trans and often show a rather specific distribution. For instance, dosage compensatory RNA Xist in mammals or Rox in *Drosophila* specifically associate with the inactive female X or the hyperactive male X chromosome in vivo (*Grimaud and Becker, 2010*; *Galupa and Heard, 2018*; *Samata and Akhtar, 2018*; *Plath et al., 2002*). RNA transcribed from pericentromeric or centromeric chromatin bind plays a key role in setting up the structure of the centromere and the clustering of distinct centromeres in interphase (*Corless et al., 2020*; *Zhu et al., 2023*). To investigate the RNA-dependent proteomic neighbourhood of distinct nuclear domains like the hyperactive X chromosome or the chromocenter, we performed AMPL-MS for MSL2 and CID in the presence or absence of RNase A (*Figure 4a* and *Supplementary files 9–12*). As previously shown, RNA depletion disrupts the centromeric domain and the hyperactive X chromosome, while maintaining the overall nuclear morphology (*Figure 4b, c* and *Figure 4—figure supplement 1*). Thanks to the AMPL-MS method we could study the effect of an RnaseA treatment on the proteomic environment of the signature proteins of chromosomal domains. As expected, neither the targeted signature factor nor proteins that mainly interact with them by protein–protein interactions such as MSL1,3 and MOF for MSL2 or Cenp-C for CID are affected by RNAase treatment. However, the proteomic composition of the characterized domains changes substantially upon RNAse treatment (*Figure 4d, f*). Not surprisingly, a large percentage of proteins that depend on the presence of RNA contain RNA-binding domains (*Figure 4e, g*), suggesting that the proximity is mediated by a direct interaction of these factors with RNA.

The APEX2 enzyme can also biotinylated RNA efficiently when provided with biotin-anilin (*Zhou et al., 2019*) as a substrate. To investigate whether specific RNAs localize to the particular chromatin domains we therefore used biotin-anilin to selectively label RNAs in proximity to MSL1 or CID. Similar to the use of biotin-phenol, biotin-anilin results in a localized labelling of RNA that corresponds to the X chromosome territory or the centromere when labelled with pA-APEX2 and anti-MSL2 or anti-CID, respectively (*Figure 5a*). We then isolated the biotinylated RNA using streptavidin beads, removed any potentially contaminating DNA by an extensive treatment with DNAase and performed an RT-PCR reaction using primers complementary to various candidate RNAs. As expected, both roxRNAs were strongly enriched in proximity to MSL2 whereas RNA derived from centromeric G2 repeats are detected exclusively in proximity to CID (*Figure 5b, c*). A control region covering a non-centromeric repeat element know to be transcribed in SL2 cells was neither found close to CID nor to MSL2 (*Figure 5c*).

Interestingly, a few common proteins are lost from both domains, including the RNA helicase MLE, which has been shown to influence dosage compensation (*Samata and Akhtar, 2018*; *Morra et al.,*

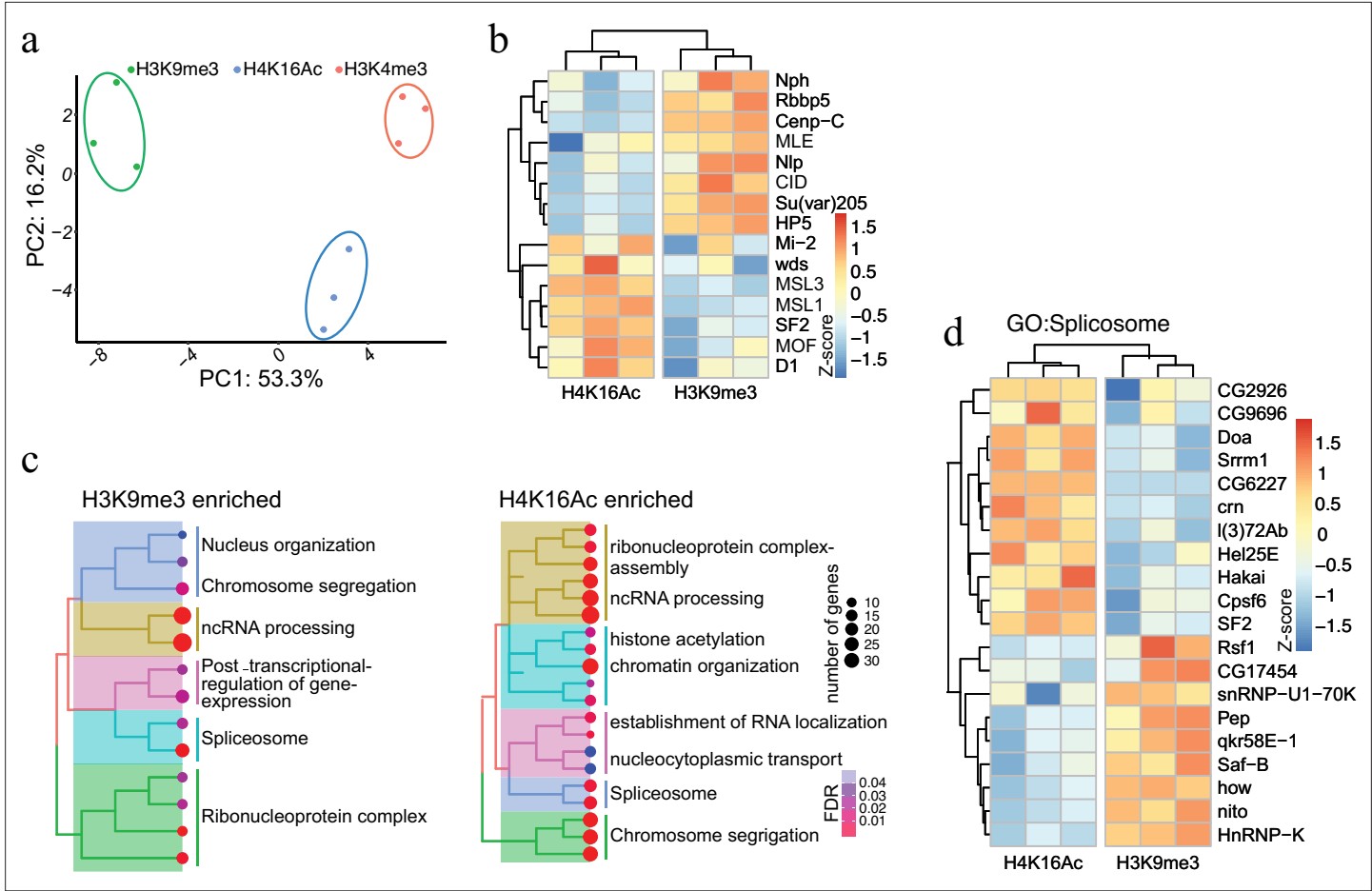

**Figure 3.** Protein associated with different and dynamic histone modifications can be effectively identified by antibody-mediated proximity labelling coupled to mass spectrometry (AMPL-MS). (**a**) Principal component analysis (PCA) based on the proteomic composition for three different histone marks: H3K9me3, H4K16ac, and H3K4me3 antibodies. (**b**) Heatmap displaying the enrichment of known H3K9me3 and H4K16ac histone modification associated proteins. The heatmap was plotted using scaled $\log_2$ raw intensities. Each column represents values obtained from three independent biological replicates. (**c**) Over representation analysis showing top 20 biological process (BP) for the significantly enriched protein associated with either H3K9me3 or H4K16ac histone modifications. The colour gradation form blue to red represents FDR (false discovery rate) and dot size represents the number of proteins found enriched in the named pathway (count). (**d**) Heatmap for protein categorized under the gene ontology (GO) term 'Spliceosome', a common GO term found for proteins associated with H3K9me3 and H4K16ac histone modification. The heatmap was plotted using scaled $\log_2$ raw intensities. Each column represents values obtained from three independent biological replicates.

The online version of this article includes the following figure supplement(s) for figure 3:

**Figure supplement 1.** Antibody-mediated proximity labelling coupled to mass spectrometry (AMPL-MS) for histone modification (related to *Figure 3*).

*2011*; *Maenner et al., 2013*) and several other nuclear processes such as splicing and heterochromatin deposition (*Cugusi et al., 2015*). The vertebrate homologue of MLE, DHX9, is an RNA–DNA helicase that has been shown to be important in the regulation of RNA–DNA hybrids (R-loops) (*Chakraborty et al., 2018*) and has been suggested to play an important role in centromere function in several organisms (*Liu et al., 2021*; *Mishra et al., 2021*; *Kabeche et al., 2018*). Based on the observation that MLE can be detected within the chromocenter of *Drosophila* polytene chromosomes (*Kotlikova et al., 2006*), a potential role for MLE for centromere formation in *Drosophila* has also been proposed. Consistently, we also observed a minor fraction of MLE close to the centromere, which was lost upon Rnase treatment (*Figure 4—figure supplement 1*). To test whether MLE is indeed involved in R-loop formation in *Drosophila* centromeres, we used novel synthetic probes that allowed us to enrich for R-loops in vivo. To do this we generated a *Drosophila* cell line that allowed us to induce the expression of GFP fused to a tandem hybrid binding domain (HBD) from RNAseH1 or a mutant version that is no longer able to bind to R-loops (HBD$_{mut}$) (*Figure 5—figure supplement 1*). Similar to what

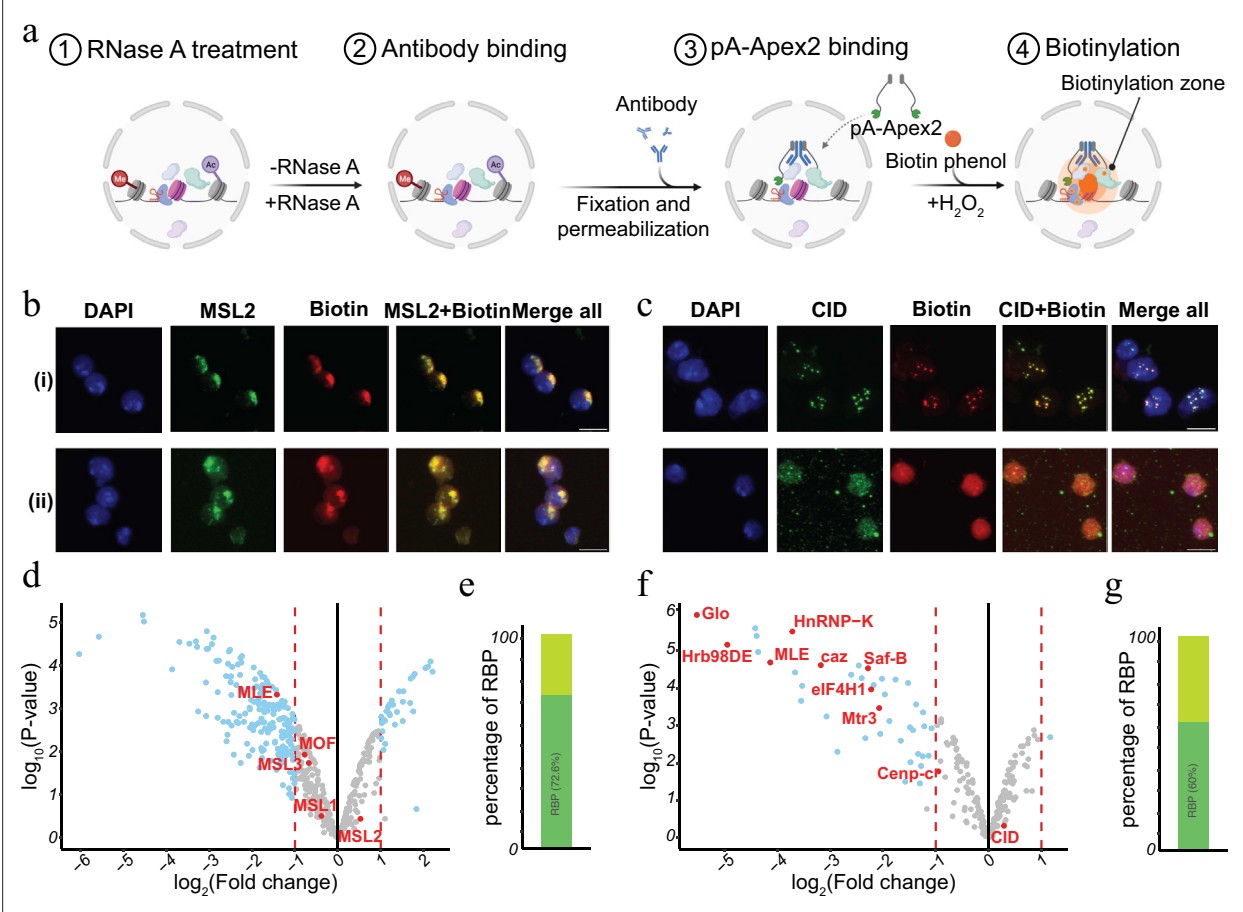

**Figure 4.** RNase treatment changes the proteomic environment of nuclear domains. (**a**) Schematic of AMPL-MS-RNase method. Isolated nuclei were treated with RNase A, fixed, permeabilized, and incubated with specific antibodies. Recombinant pA-Apex2 enzyme binds to the antibody and biotinylates associated proximal proteins upon the addition of $H_2O_2$ and biotin-phenol. Created with BioRender.com. (**b**) Immunofluorescence of the X-chromosome-bound by an MSL2 antibody (in green) and biotinylated proteins after biotinylation by pA-Apex2 in control (**i**) and RNase A-treated samples (**ii**). Nuclei was stained by DAPI (in blue). Scale bars represent 5 μm. (**c**) Immunofluorescence of the centromer bound by an anti-CID antibody (in green) and biotinylated proteins after biotinylation by pA-Apex2 in control (**i**) and RNase A-treated samples (**ii**). Nuclei were stained by DAPI (in blue). Scale bars represent 5 μm. Volcano plot of proteins identified by AMPL-MS-RNase using an anti-MSL2 (**d**) or an anti-CID antibody (**f**). The bait protein is highlighted in red, along with components of the *Drosophila* dosage compensation complex. The x-axis represents the $\log_2$ fold change and the y-axis represents $-\log_{10}$ p-value comparing three control MSL-2 AMPL-MS replicates with three MSL2-RNase A-treated antibody-mediated proximity labelling coupled to mass spectrometry (AMPL-MS) replicates (paired). The significantly enriched proteins (LFC >1 and $p_{adj} \leq 0.01$) are highlighted in blue. Percentage of RNA-sensitive proteins in proximity to MSL2 (**e**) or CID (**f**) containing known RNA-binding domains (RBP).

The online version of this article includes the following figure supplement(s) for figure 4:

**Figure supplement 1.** Maleless (MLE) associated with centromere in an RNA-dependent manner (related to *Figure 4*).

has been shown in other species (*Chakraborty et al., 2018*; *Liu et al., 2021*; *Kabeche et al., 2018*; *Racca et al., 2021*) using this reporter we detect R-loops at the *Drosophila* centromere (*Figure 5d*). DHX9 recognizes and suppresses physiological R-loops at transcription termination regions and CPT-induced R-loops (*Cristini et al., 2018*). However, in another study DHX9 has been shown to suppress toxic R-loops as well as promote regulatory R-loops, by means of its ability to unwind secondary DNA and RNA structures (*Chakraborty et al., 2018*). Likewise, *Drosophila* centromeres have been shown to be rich in sequences that could potentially form non-canonical secondary structures (*Patchigolla and Mellone, 2022*). These unusual DNA structures may help recruiting CID-specific chaperones such as HJURP in tetrapods (*Kato et al., 2007*). Such structures form on single-stranded DNA and may therefore be stabilized through the formation of R-loops. Based on our results, we therefore speculate

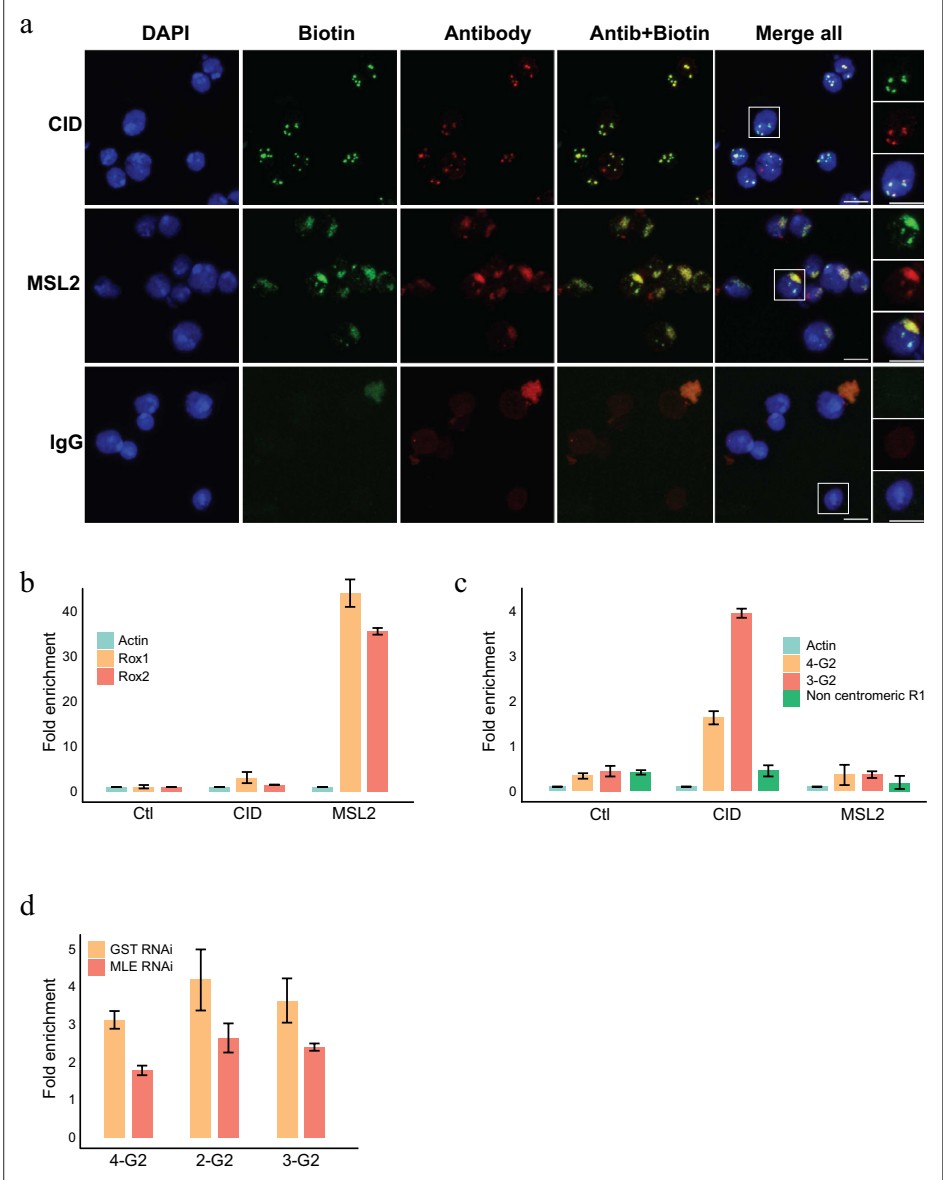

**Figure 5.** RNA labelling using antibody-mediated proximity labelling (AMPL) to study the RNA composition of chromatin domains. (**a**) Immunofluorescence microscopy of centromeres and hyperactive X chromosome using CID and MSL2 antibody, respectively (in green), and the corresponding proximity RNA labelling after biotinylation by pA-Apex2 (in red). Nuclear DNA was stained by DAPI (in blue). IgG was used as antibody control. Scale bars represent 5 µm. (**b**) RT-qPCR analysis of enriched RNA in proximity to X chromosome. The PCR analysis is showing a specific enrichment of two long non-coding RNA *Rox1* and *Rox2* which are known to associate with X chromosome and actin as control. Relative abundance is calculated following $2^{-\Delta\Delta Ct}$ method. Data are the mean of three replicates ±1 standard deviation (SD). (**c**) RT-qPCR analysis of enriched RNA in proximity to centromere. The PCR analysis is showing specific enrichment of RNA originates from centromeres 3 and 4 of SL2 cells. R1 is a control for non-centromeric transcript. Data are the mean of three replicates ±1 SD. (**d**) Enriched R-loop-ChIP-qPCR using hybrid binding domain (HBD) probe. The graph shows enrichment of R-loop in centromere for chromosomes 2–4 in control cells (gst RNAi) in comparison to MLE knock down cells. The enrichment is calculated relative to the input and is normalized by the *Sdr* promoter region as a non-centromeric control. Data are the mean of three replicates ±1 SD.

The online version of this article includes the following source data and figure supplement(s) for figure 5:

**Figure supplement 1.** R-loop associated with centromere (related to *Figure 5*).

**Figure supplement 1—source data 1.** Raw image files of the *Figure 5—figure supplement 1*.

**Figure supplement 1—source data 2.** Labelled raw image files of the *Figure 5—figure supplement 1*.

that MLE promotes the formation of R-loops by resolving secondary structures that will form in the centromeric RNA (*Chakraborty and Grosse, 2011*; *Figure 5—figure supplement 1*). Such R-loops will then drive the generation of unusual DNA structures that in turn might facilitate CID recruitment. The centromere-associated RNA could then help stabilizing MLE at the centromere thereby resolving the secondary structures and allowing CID containing nucleosomes to assemble. Consistent with this hypothesis we find a substantial reduction of the centromeric R-loops upon knocking down MLE (*Figure 5d*). This model is further supported by recent findings that a removal of R-loop causes the loss of Cenp-A from the mammalian centromere (*Kitagawa et al., 2023*) and that neocentromere formed on ectopic sites are often associated with the presence of R-loops (*Arunkumar et al., 2022*).

In summary, our data show that the versatility of AMPL-MS allows us to characterize membrane less domains in the nucleus at an unprecedented molecular level and therefore identify systematic changes in these organelles upon challenge.

## Methods

### Cell culture

*Drosophila* S2DGRC was purchased from the *Drosophila* Genomics Research Center (https://dgrc.bio.indiana.edu) a subclone of them (L2–4) was a gift from Patrick Heun. Both lines have been verified for identity by long read sequencing. Cells were grown in Schneider medium supplemented with 10% fetal calf serum, penicillin and streptomycin at 26°C. Both cell lines have been tested negative for Mycoplasma.

### Cloning

To construct a bacterial expression vector for protein-A-Apex2-6xHis fusion protein (pk19pA-Apex2), the MNase sequence in the pk19pAMNase vector (pk19pAMNase was a gift from Ulrich Laemmli, Addgene #86973) was replaced by the Apex2-6xHis sequence. Apex2 was amplified from the plasmid Apex-2NLS_pMT-Hygro (*Kochanova et al., 2020*) and inserted in the pK19 vector between EcoRI and BamHI restriction enzymes using the primers 'pProA-Apex2-3Gly-6His FW' and 'pProA-Apex2-3Gly-6His RE'. Cloning was performed with In-Fusion cloning kit (Clontech). To generate the HBD expression construct for *Drosophila* (i.e., pMT-EGFP-2xHBD-3xFLAG or the mutant pMT-EGFP-2xHBD WKK-3xFLAG). The 'EGFP-2xHBD-3xFLAG' or 'EGFP-2xHBD(WKK)-3xFLAG' was amplified from pcDNA4_TO_EGFP_2xHBD-3xFLAG or pcDNA4_TO_EGFP_2xHBD(WKK)-3xFLAG and inserted into Not1 and Bgl2 digested 'pMT-Hygromycin' expression plasmid backbone using In-Fusion cloning kit (Clontech). To generate a stable cell line, 3–4 millions of cells were transfected with 2 µg of plasmid mixed with XtremeGENE HP (Roche) transfection reagent according to the manufacturer's instructions. After transfection, cells were selected for 3 weeks with Hygromycin B (Invitrogen) at 100 µg/ml, and were selected on Hygromycin during further culture and experiments. Optional induction of cell lines with 250 µM CuSO$_4$ was performed 12–24 hr before experiments. Western blot was performed to verify the production of the HBD probes.

### Expression and purification of pA-Apex2 enzyme

The bacterial expression plasmid (pk19pA-Apex2-6xHis) was transformed into Bl21DE3 Gold (Stratagene# 230132) according to the manufacturer's instruction. Transformed colonies were grown overnight and re-cultured in a fresh 1 l of Luria-Bertani (LB) medium until the optical density (OD) reaches to a value 0.6. The protein expression was then induced by adding Isopropyl β-d-1-thiogalactopyranoside (IPTG 0.25 mM) and cultured for additional 3 hr. Cells were harvested and resuspended in 40 ml of lysis buffer 25 mM Tris pH: 7.2, 10% glycerol, 150 mM NaCl, 0.1% Octylphenolpolyethoxyethanol (IGEPAL), 1 mM Dithiothreitol (DTT), 10 mM imidazole, supplemented with cOmplete EDTA-free Protease Inhibitor Cocktail (Roche)). Cells were at first lysed with lysozyme and then sonicated. The cell lysate was centrifuged at 45,000 rpm for 30 min at 4°C. The cleared lysate was then incubated with 1 ml of TALON metal affinity (Takara #Z5504N) resin according to the manufacturer's instruction. Lysate with the beads were passed through a column (Econo-Pack Disposable Chromatography Columns, 10 ml, Bio-Rad) by gravity flow. The column was then washed three times with lysis buffer, and twice with wash buffer (25 mM Tris pH: 7.2, 10% glycerol, 150 mM Nacl, 0.1% IGEPAL, 1 mM DTT, 20 mM imidazole, supplemented with cOmplete EDTA-free Protease Inhibitor Cocktail (Roche)). The

enzyme ProtA-Apex2 was eluted with 5 ml of wash buffer containing 200 mM imidazol, collected in 1 ml fractions. A small aliquot of the collected fractions was loaded on to a sodium dodecyl sulphate–polyacrylamide gel electrophoresis (SDS–PAGE) gel and stained with InstantBlue Coomassie protein stain (abcam #ab119211) according to the manufacturer's instruction. Fractions containing pA-Apex2 were pulled together and dialysed overnight in dialysis buffer (25 mM Tris pH: 7.2, 10% glycerol, 150 mM NaCl, 0.1% IGEPAL, 1 mM DTT, and protease inhibitors). The dialysed protein solution was then concentrated using Amicon Ultra-4 Centrifugal Filter Units 30K (Millipore, MA #UFC803024) according to the manufacturer's protocol. The enzyme was snap-frozen and stored at $-80°C$ in storage buffer (25 mM Tris pH: 8, 30% glycerol, 150 mM NaCl, 0.1% IGEPAL, 1 mM DTT, 2 mg/ml of biotin-free Bovine serum albumin (BSA) (Roth #0163.2), supplemented with cOmplete EDTA-free Protease Inhibitor Cocktail (Roche). The concentration of pA-Apex2 was determined using BSA standards.

## In vitro activity assay of pA-Apex2

Cells were harvested, washed 2× in phosphate-buffered saline (PBS), and resuspended in hypotonic buffer (20 mM 4-(2-hydroxyethyl)-1-piperazineethanesulfonic acid (HEPES) pH 7.9, 20 mM NaCl, 5 mM $MgCl_2$, 1 mM Phenylmethylsulfonyl fluoride (PMSF), 1 mM DTT, supplemented with cOmplete EDTA-free Protease Inhibitor Cocktail (Roche)) and incubated on ice for 10 min. Subsequently, cells were dounced with a $26^1/_2$ G needle and again incubated on ice for 10 min. Nuclei were pelleted at 500 × $g$, 5 min, 4°C and lysed in hypotonic buffer supplemented with 0.5% IGEPAL CA-630 (Sigma). To the lysate, Benzonase (Millipore) was added and rotated at 4°C for 1 hr. Subsequently, NaCl concentration was raised to 300 mM through addition of 5 M NaCl to lysate and rotated at 4°C for another 30 min. NaCl concentration was lowered back to 150 mM through addition of hypotonic buffer supplemented with 0.5% IGEPAL CA630 and lysate cleared by centrifugation (15,000 × $g$, 15 min, 4°C). For experiments shown in *Supplementary file 1b, c* an aliquot of the sample was incubated with 5 µM pA-Apex2 and 500 µM biotin-phenol (BP) for 1 min. The biotinylation reaction was triggered by adding $H_2O_2$ to a final concentration of 2 mM for 1 min. The reaction was stopped by adding quenching solution. As a negative control similar reaction was performed either in absence of $H_2O_2$, BP, or $H_2O_2$ and BP.

## Western blotting

Biotinylated extract from in vitro assay or biotin immunoprecipitated streptavidin bead-bound proteins were eluted in Lemmli Buffer at 95°C and resolved on an SDS–PAGE gel (Serva #43264.01). Proteins were then transferred to a polyvinylidene difluoride (PVDF) membrane using a Trans-Blot Turbo Transfer System (Bio-Rad) according to the manufacturer's instructions. Membranes were blocked in blocking solution (2% biotin-free BSA in PBS) in a shaker for 1 hr at room temperature (RT). Membranes were then incubated with HRP–streptavidin (BioLegend #405210) (1:2000) in 2% biotin-free BSA in PBST (PBS with 0.1% Tween) for 1 hr in RT. The membranes were developed using Clarity Western ECL Substrate (Bio-Rad #170-5061) and imaged using ChemiDoc Touch Imaging Syatem (Bio-Rad).

## AMPL-MS proximity biotinylation and immunoprecipitation

For AMPL-MS experiments cells were grown in T75 flasks (Greiner) to a density of $1 × 10^7$ cells/ml. For each experiment, $2 × 10^7$ cells were used, therefore $32 × 10^7$ for a set of experiment (i.e., including experiment, antibody control, biotin-phenol control, and $H_2O_2$ control). The cells were washed with 20 ml cold PBS in a 50-ml tube and centrifuged (Thermo Scientific, Heraeus, Multifuge X3R) at 250 × $g$, 4°C for 5 min. Next, for the nuclear isolation the cell pellet was resuspended in three packed cell volumes of nuclear isolation buffer (NIB) (20 mM Tris pH 7.6, 10 mM KCl, 2.5 mM $MgCl_2$, 0.5 mM Ethylenediaminetetraacetic acid (EDTA), cOmplete Protease inhibitors), incubate on ice for 10 min. Following the incubation, the cell suspension was supplemented with NP40 to a final concentration of 1% and pass through a 20 G needed. The nuclei were spun down at 500 × $g$ for 5 min, and washed with NIB with 0.1% NP40. The nuclei were then fixed using in AMPL-MS assay buffer (20 mM HEPES pH 7.5, 150 mM NaCl, 2.5 mM $MgCl_2$, 0.5 mM EDTA, cOmplete Protease inhibitor, MG132, 0.5 mM Spermidine) containing 3.7% vol/vol paraformaldehyde and incubated on a rotating wheel for 10 min in RT. The reaction was quenched by adding 1/20 volume of 2.5 M glycine. Subsequently, the nuclei were washed twice with AMPL-MS assay buffer and briefly treated with $H_2O_2$ (5 mM) and quickly washed with assay buffer. Following the treatment, the nuclei were permeabilized with 0.25% Triton X-100 for 6 min on ice and washed in assay buffer supplemented with 1% BSA (biotin-free BSA).

The nuclei were then blocked with Image-iT FX Signal Enhancer (Invitrogen #I36933) for 45 min on a rotating wheel at RT. The nuclei were then quickly washed with assay buffer and resuspended in antibody incubation buffer (assay buffer with 5% NGS (Jackson ImmunoResearch), 0.02% digitonin) and split equally into four 0.5 ml low protein-binding tubes. The respective antibodies were added (2.5 μg/reaction) and incubated overnight in the cold room on a rotating wheel. To remove unbound antibody, following day, the nuclei were washed thrice with wash buffer (assay buffer with 0.1% Tween 20). Next, the nuclei were incubated with 2.5 μg of pA-Apex2 in antibody incubation buffer for 2 hr in the cold room and another 1 hr in RT on a rotating wheel. After the incubation the nuclei were washed thrice with wash buffer to remove the unbound enzyme. Following the washes biotinylation was performed. The nuclei were first incubated with biotin-phenol (500 μM) in assay buffer for 20 min in RT in the dark, then $H_2O_2$ (1 mM) was added for 2 min to start the biotinylation reaction. The quenching buffer (Trolox 5 mM, sodium sscorbate 10 mM, sodium azide 10 mM) was added to stop the reaction. The nuclei were washed twice with wash buffer with quenching solution. A small amount of the nuclei was saved for immunofluorescence. The rest of the nuclei were subjected to lysis and decrosslinking by heating at 99°C on a Thermo shaker (800 rpm) for 1 hr in nuclear lysis buffer (100 μl PBST (PBS with 0.1% Tween 20) 30 μl of 10% SDS and 20 μl of 10% sodium deoxycholate). To adjust the concentration of the detergent for immune precipitation the volume of the samples was made up to 1 ml with PBST. Subsequently the samples were treater with 100 U of Benzonase (Milipore #1.01654.0001) and spun down at 14,000 rpm for 20 min at 4°C. A small fraction (5%) of the supernatant was kept aside as input. The rest of the lysate was incubated with 50 μl of precleaned streptavidin beads (Invitrogen, Dynabeads M-280 Streptavidin #11206D) at RT for 2 hr. The beads were then washed twice with PBST, twice with PBST + 1 M NaCl, twice with PBS. The beads were transferred to a new tube and washed thrice with 50 mM ammonium bicarbonate. A small fraction (5%) of the beads were saved for immunoblot analysis. The rest of the beads were subjected to on-bead digestion for mass spectrometry.

## AMPL-RNA labelling and enrichment

To label RNA using antibody-mediated pA-Apex2 the proximity biotinylation was performed as above with slight modification. Unless otherwise noted all the buffers used during RNA labelling and isolation was performed in RNase-free condition. Briefly, $8 \times 10^7$ cells were washed with PBS followed by fixation with 0.1% formaldehyde (FA) for 10 min in RT. The cells were then incubated with respective antibody and pA-Apex2 as above. The biotinylation reaction was performed using biotin-aniline (Iris Biotech, #LS-3970). After the biotin labelling, the nuclei were isolated as above. The isolated nuclei were lysed and decrosslinked in 200 μl of 1× SDS solubilization buffer 0.5% SDS, 1 mM EDTA, 20 mM Tris–Cl (pH 7.5) (+RNase inhibitor) with Proteanase K (100–200 μg/1 × 10$^7$ cells) at 42°C for 1 hr, followed by 55°C for 1 hr. The RNA was purified using TRIzol (Zymo research). The isolated RNA was treated with RNAse-free DNAse I (NEB), followed by phenol–chloroform extraction. To enrich biotinylated RNA C1 streptavidin magnetic beads (Invitrogen) were used (10 μl beads per 25 μg of RNA). The beads were cleaned three times with wash buffer (100 mM Tris–HCl (pH 7.4), 10 mM EDTA, 1 M NaCl, 0.1% Tween 20), followed by three times with 0.1 M NaOH, and three times with 0.05 M NaCl and 1 time with 0.1 M NaCl. The beads were blocked in 1× Denhardt's solution (Sigma #750018) with yeast tRNA 1 μg/μl for 2 hr in RT. After the incubation the beads were washed with 1× Denhardt's solution and 100 mM NaCl. The beads were resuspended in 100 mM NaCl. The blocked beads were then mixed with 50 μg of RNA in Diethyl pyrocarbonate (DEPC) treated water to a 50 mM NaCl final concentration and incubated in cold room for 2 hr on a rotating wheel, followed by three times washes using wash buffer. The beads were then resuspended in 54 μl of RNAse-free water. A 3× proteinase digestion buffer was made 1.1 ml of buffer containing 330 μl of 10× PBS pH 7.4 (Ambion), 330 μl 20% N-lauryl sarcosine sodium solution (Sigma-Aldrich), 66 μl 0.5 M EDTA (Ambion), 16.5 μl 1 M DTT (Thermo Fischer). 33 μl of the 3× proteinase buffer was added to the beads along with 10 μl of Proteinase K (20 mg/ml, Ambion) and 3 μl Ribolock RNase inhibitor. The beads were then incubated at 42°C for 1 hr, followed by 55°C for 1 hr. The RNA was then purified with RNA clean and concentrator kit (Zymo Research). The enriched and purified RNA was then used for reverse transcription PCR (RT-PCR) for quantification . Briefly, the RNA was treated with RNase-free DNAse I (ROCH) followed by first strand cDNA synthesis using SuperScript III (Invitrogen #18080-051) according to the manufacturer's protocol using random hexamers. The cDNA was treated with RNase H and used for further analysis.

## Immunofluorescent staining

Immunofluorescent experiments were performed as described previously (*Kochanova et al., 2020*) with minor modifications. Briefly, the nuclei from AMPL-MS assays were adhered on a poly-L-lysine-coated glass coverslip, washed with PBST with quenching solution. After washes, the nuclei were incubated with secondary antibodies for respective bait coupled to Alexa Fluor 488, and anti-biotin antibody [streptavidin, Alexa Fluor 555 #S32355 (1:600)] for 1 hr at RT. Slides were again washed 3 × 5 min with 0.1% Triton X-100/PBS and incubated with DAPI for 3 min. Excess DAPI was washed off with 0.1% Triton X-100/PBS for 5 min and samples mounted with VECTASHIELD (Vector Labs). Images were acquired using Leica TCS SP8 Confocal Microscope, processed and quantified using Fiji (*Schindelin et al., 2012*). The colocalization analysis was performed using the EzColocalization package in Fiji.

## Mass spectrometry

Mass spectrometry-based proteomic experiments were performed as described previously (*Kochanova et al., 2020*) with minor modifications. Briefly, beads were washed three times with 50 mM $NH_4HCO_3$ and incubated with 10 ng/μl trypsin in 1 M urea 50 mM $NH_4HCO_3$ for 30 min, washed with 50 mM $NH_4HCO_3$ and the supernatant digested overnight in presence of 1 mM DTT. Digested peptides were alkylated and desalted prior to LC–MS analysis. The peptide mixtures were subjected to nanoRP-LC–MS/MS analysis on an Ultimate 3000 nano chromatography system coupled to a Qexactive HF or a Orbitrap Exploris-480 mass spectrometer (both Thermo Fisher Scientific) in two to four technical replicates (5 μl each).

## RNAi

RNAi against a target gene was performed as in *Müller et al., 2020*. Briefly, 1 million cells were seeded in a 6-well plate and grown overnight; next day the medium was removed and a total of 10 μg of dsRNA (5 μg each as two different siRNAs were used to increase the knock down efficiency) in 1 ml serum-free Schneider medium was added. Cells were gently rocked on a platform for 10 min at RT and left for additional 50 min at 26°C. Afterward 2 ml of medium was added. On day 5, cells were resuspended and counted. $2 \times 10^7$ cells per RNAi condition were transferred to a 75-cm$^2$ flask for a second round of dsRNA treatment (80 μg dsRNA/flask/RNAi target) in 8 ml serum-free media and incubated as mentioned above. At day 10, cells were counted and processed for R-loop-ChIP.

## R-loop-ChIP

R-loop-ChIP experiments were performed on cells expressing the either 'EGFP-2xHBD-3xFLAG' or HBD mutant 'EGFP-2xHBD(WKK)-3xFLAG' protein by using standard ChIP protocol as in *Lukacs et al., 2021* with modifications. Briefly, 5–7 × 10$^7$ cells after RNAi treatment were harvested and crosslinked with 1% FA for 10 min in RT. The reaction was stopped by adding glycine at 125 mM final concentration and incubating for 10 min on ice. Cells were washed twice in PBS before subsequent steps. For nuclei isolation, cells were incubated in nuclear isolation buffer followed by addition of 1% final volume of NP40. After nuclear isolation cells were resuspended in PBS supplemented with 0.5% (vol/vol) Triton X-100 and cOmplete EDTA-free Protease Inhibitor Cocktail (PI; Roche), volume was adjusted to 7 × 10$^7$ cells/ml and cells incubated for 15 min at 4°C with end-over-end rotation. Nuclei were collected by centrifuging at 4°C for 10 min at 2000 × *g* and washed once in PBS. For chromatin fragmentation, nuclei were spun down at 4°C for 10 min at 2000 × *g*, resuspended in RIPA buffer (10 mM Tris/HCl pH 8.0, 140 mM NaCl, 1 mM EDTA, 1% (vol/vol) Triton X-100, 0.1% (vol/vol) SDS, 0.1% (vol/vol) sodium deoxycholate) supplemented with PI. The chromatin was sheared to 250–600 bp in size by sonication with Covaris AFA S220 using 12 × 12 tubes at 100 W peak incident power, 20% duty factor and 200 cycles per burst for 12 min at 5°C. The samples were precleaned using protein A/G Sepharose. 5% chromatin fragment was saved as input and the remaining was incubated with with anti-FLAG antibody overnight at 4°C. Next morning protein-A/G Sepharose beads were added to isolate the antibody-bound chromatin. Beads were sequentially washed three times with wash buffer-I (20 mM Tris–HCl pH 8.0, 150 mM NaCl, 1% Triton X-100, 0.1% SDS, 2 mM EDTA, and 1× protease inhibitor cocktail), three times with wash buffer-II (20 mM Tris–HCl pH 8.0, 500 mM NaCl, 1% Triton X-100, 0.1% SDS, 2 mM EDTA, and 1× protease inhibitor cocktail), once with wash buffer -III (10 mM Tris–HCl pH 8.0, 250 mM LiCl, 1% NP40, 1% deoxycholate, 1 mM EDTA, and 1×

protease inhibitor cocktail) and once with TE buffer (10 mM Tris–HCl pH 8.0 and 1 mM EDTA). After the washes, the beads along with the input samples were incubated in 100 µl of TE buffer with 50 µg/ml of RNase A at 37°C for 30 min. After RNase treatment, the samples were adjusted to 0.5% SDS, 0.5 mg/ml of Proteinase K and incubated overnight at 65°C, 1400 rpm on a thermomixer to decross-link. The DNA sample was purified using AMpure XP Beads (Beckmann Coulter). The recovered DNA was diluted to 1:10 ratio and subjected to qPCR analysis.

### Database search

MaxQuant 1.6.1.476 (*Tyanova et al., 2016*) was used to identify proteins and quantify by LFQ with the following parameters: Database, dmel-all-translation-r6.08.fasta (Flybase); MS tolerance, 10 ppm; MS/MS tol, 20 ppm; Peptide false discovery rate (FDR), 0.1; Protein FDR, 0.01; Min. peptide length, 5; Variable modifications, Oxidation (M); Fixed modifications, Carbamidomethyl (C); Peptides for protein quantitation, razor and unique; Min. peptides, 1; Min. ratio count, 2. Match-between-runs (MBR) option was selected. Technical replicates were assigned to one experiment (biological replicate). Experimental and control samples (treated with biotin-phenol and DMSO, respectively) were loaded into the same MaxQuant run. Samples from different cell lines and time points were run separately.

### Data analysis

The output files from MaxQuant (proteinGroups.txt) were analysed in R environment. Data were filtered such that proteins that are present in two of the three replicates of at least one condition were taken for the analysis. Following filtering, MinProb imputation algorithm with $q = 0.01$ was performed to impute the missing values and limma based differential expression analysis was carried out. Proteins were considered significant if the LFC >1 and FDR ≦0.05. Over representation analysis was performed using the enrichGO function from clusterProfiler (*Lyon et al., 2021*) package v3.12.0 by taking the significant proteins (FDR 0.05 cut-off for predicted GO terms) and without a background set. Corresponding GO plots were also generated with R environment. For the tree plot, GO terms were clustered based on GO semantic similarity. Based on the representation in each cluster, a summarized GO term was the written. All code is freely available at https://github.com/anuroopv/RAmP (copy archived at *VenkateswaranVenkatasubramani, 2024*).

### Histone PTM analysis

To look at histone modification associated with different chromatin domains as shown in *Figure 2f* and *Figure 2—figure supplement 1*, we performed AMPL-MS assay as explained above and performed histone PTM analysis as described previously (*Völker-Albert et al., 2018*) with minor modifications. In brief, the biotinylated proteins were immunoprecipitated using streptavidin beads as earlier and eluted in 1× SDS lysis buffer. The proteins were then resolved on a precast SERVAGel TG PRiME 4–20% (SERVA Electrophoresis GmbH) gel and stained with InstantBlue Coomassie Protein Stain. Protein bands which correspond to the histones (expected between 11 and 17 kDa) were cut out and destained. Following destaining, in-gel histone acylation using propionic anhydride and digestion with trypsin [MS Grade Pierce Trypsin Protease (Thermo Scientific)] were performed. The histone peptides were then extracted and cleaned using a C8 Stagetip (*Rappsilber et al., 2007*) before mass spectrometry analysis. MS1 peak integration was performed using Skyline software (*Pino et al., 2020*) and relative abundances of H3 and H4 peptides were calculated using R.

### Plots and statistical analysis

All statistical analysis was performed in R environment, except for *Figures 1c and 2b*. Plots and graphs were generated in R environment, except for *Figures 1c, 2b, g, and 3f, e*. All schematic figures were created with BioRender.com.

### Data sources

The datasets produced in this study are available in the ProteomeXchange Consortium via the PRIDE (*Perez-Riverol et al., 2022*) partner repository with the identifiers: PXD044295 (Proteomics) and PXD044296 (Histone PTMs).

## Acknowledgements

We would like to thank Catherine Regnard and Viola Gilardino for their advice on R-loop analysis the members of the Imhof lab and Becker department for their discussion and excellent suggestions. In addition, we thank Markus Hohle from QBM for his constant support. All schematic figures were created with BioRender.com. The project was funded by a grant of the Volkswagenstiftung to SK (grant number 97131) and grants from the DFG to AI (grant numbers 213249687 (CRC1064), 325871075 (CRC1309), and 419067076 (SPP2191)).

## Additional information

### Funding

| Funder | Grant reference number | Author |
|---|---|---|
| Deutsche Forschungsgemeinschaft | 419067076 | Rupam Choudhury |
| Deutsche Forschungsgemeinschaft | 213249687 | Marco Borsò |
| Deutsche Forschungsgemeinschaft | 325871075 | Rupam Choudhury |
| Deutsche Forschungsgemeinschaft | QBM | Anuroop Venkateswaran Venkatasubramani |
| Volkswagen Foundation | 97131 | Celeste Franconi Sarah Kinkley |

The funders had no role in study design, data collection, and interpretation, or the decision to submit the work for publication.

### Author contributions

Rupam Choudhury, Conceptualization, Investigation, Visualization, Methodology, Writing - original draft; Anuroop Venkateswaran Venkatasubramani, Visualization, Methodology, Writing - review and editing; Jie Hua, Investigation, Writing - review and editing; Marco Borsò, Data curation, Formal analysis, Investigation, Visualization; Celeste Franconi, Sarah Kinkley, Resources, Methodology; Ignasi Forné, Formal analysis, Supervision, Investigation, Visualization, Methodology, Project administration, Writing - review and editing; Axel Imhof, Conceptualization, Data curation, Supervision, Funding acquisition, Writing - original draft, Project administration, Writing - review and editing

### Author ORCIDs

Rupam Choudhury http://orcid.org/0009-0005-2669-7218
Anuroop Venkateswaran Venkatasubramani http://orcid.org/0000-0003-2119-8741
Jie Hua http://orcid.org/0000-0003-4910-0945
Marco Borsò http://orcid.org/0000-0001-7467-7960
Sarah Kinkley http://orcid.org/0000-0003-4997-4749
Ignasi Forné http://orcid.org/0000-0003-0309-907X
Axel Imhof http://orcid.org/0000-0003-2993-8249

### Decision letter and Author response

Decision letter https://doi.org/10.7554/eLife.95718.sa1
Author response https://doi.org/10.7554/eLife.95718.sa2

## Additional files

### Supplementary files

• Supplementary file 1. CID-vs-AB: comparison of AMPL-MS proximity proteomics between CID antibody vs non-specific antibody as plotted in volcano plot *Figure 1d*.

• Supplementary file 2. ORA_CID-vs-AB: over representation analysis of protein identified in proximity to the centromeric chromatin domain (CID) as plotted in *Figure 1f*.

• Supplementary file 3. MSL2-vs-CID: comparison of AMPL-MS proximity proteomics between CID

antibody vs Msl2 antibody as plotted in volcano plot *Figure 2c*.

• Supplementary file 4. ORA_MSL2-vs-CID: over representation analysis of protein identified in proximity to the centromeric chromatin domain (CID) vs hyperactive X chromosome (MSL2) as plotted in *Figure 2e, f*.

• Supplementary file 5. MSL2-vs-AB: comparison of AMPL-MS proximity proteomics between Msl2 antibody vs non-specific antibody.

• Supplementary file 6. AMPL-MS histone MS: analysis of histone modification associated with centromeric (CID) or hyperactive X chromosome (Msl2) as in *Figure 2g*.

• Supplementary file 7. Histone PTMs: comparison of proteomics data obtained for proteins associated with histone post-translational modifications (H3K9me3, H3K4me3, and H4K16Ac) as in *Figure 3a,b,d* and *Figure 3—figure supplement 1*.

• Supplementary file 8. ORA_HistonePTMs: over representation analysis of protein found to be enriched in proximity to the H3K9me3 and H4K16Ac modifications as plotted in *Figure 3c*.

• Supplementary file 9. MSL2-RNase-vs-MSL2: a comparison of protein associated with hyperactiv X chromosome with and without RNAse-A treatment as plotted in *Figure 4d*.

• Supplementary file 10. CID-RNase-vs-CID: a comparison of protein associated with centromere with and without RNAse-A treatment as plotted in *Figure 4f*.

• Supplementary file 11. CID_MSl2 RNase protein list: a list of protein found to be associated with centromere and hyperactive X chromosome.

• Supplementary file 12. Primer list: a list of primers used in the study.

• MDAR checklist

### Data availability

The datasets produced in this study are available in the ProteomeXchange Consortium via the PRIDE 64 partner repository with the identifiers: PXD044295 (Proteomics) and PXD044296 (Histone PTMs).

The following datasets were generated:

The following dataset was generated:

| Author(s) | Year | Dataset title | Dataset URL | Database and Identifier |
|---|---|---|---|---|
| Imhof A | 2024 | Proximity proteomics using AMPL-MS coupled to analysis for histone modification | https://www.ebi.ac.uk/pride/archive/projects/PXD044296 | PRIDE, PXD044296 |
| Imhof A | 2024 | Proximity proteomics using AMPL-MS | https://www.ebi.ac.uk/pride/archive/projects/PXD044295 | PRIDE, PXD044295 |

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
