## [Editor Report]

This important work features an innovative proximity labeling approach to the identification of proteins enriched in distinct types of chromatin domains. The work further shows that selective protein interactions are RNA dependent. This compelling evidence may fundamentally advance our understanding of chromosome domains and the role of RNA in organizing them.

---

## [Decision Letter]

[Editors' note: this paper was reviewed by Review Commons.]

---

## [Author Response]

1. General Statements

We considered the Reviewers’ comments with all authors and feel that we will be able to address most of their concerns within the next two months. All four Reviewers appreciate the quality of our data but suggest that we should focus more on the biological findings rather than on the technology we use. While we agree with the comments of the Reviewers that the general concept of antibody-based proximity biotinylation we used to characterize nuclear domains is not entirely novel, we would still like to point out that our method results in a much higher sensitivity and selectivity compared to similar approaches published before. The use of HRP coupled secondary antibodies as suggested by Reviewer2 results in a much higher background when used for nuclear domains and the use of BirA suffers from a lower biotinylation efficiency. We will try to put more emphasis on this aspect of the method in a revised manuscript.

2. Description of the planned revisions

As the use of APEX2 also enables us to simultaneously characterize the RNA molecules in proximity to the targeted protein by using biotin-aniline as a substrate, we decided to investigate the role of RNA in the organization of nuclear domains in more detail. We have used the selective enrichment of Rox1 and the Rox2 RNA to the Msl2 domain as proof-of concept and are currently sequencing the RNA in proximity to the centromere. I have attached the data for the Rox RNAs below to demonstrate the general feasibility of the method (see figure attached). In the revised manuscript we also plan to elaborate on the functional role of our finding that Mle is found in proximity to the *Drosophila* centromere in an RNA dependent manner. Considering that the formation of RNA-DNA hybrids (R-Loops) has been shown to severely harm genome integrity, we wondered whether Mle might function to resolve the formation of R-loops at centromeric regions. In fact, we can detect an increased formation of centromere proximal R-loops upon knockdown of Mle, which coincide with a substantial widening of the centromeric domain. These results underscore a general importance of R-loops removal for the organization of centromeric domains in particular.

3. Description of the revisions that have already been incorporated in the transferred manuscript

We have not yet incorporated any new experiments in the transferred manuscript.

Detailed response to reviewers

Reviewer #1The vast majority the experimental results, statistical analysis, and conclusions drawn by the authors appear sound and are described in way that should allow reproduction (however, see my comments below for some suggestions for minor improvements). The authors rigorously test their method, using the *Drosophila* chromodomain as 'playground', before applying it to other chromosomal areas and histone variants/modifications. Besides providing proteomes of the targeted nuclear subcompartments, they show that RNase treatment of the cells radically changes the proteome(s) and conclude a role for RNA in the integrity of the corresponding compartments.

We thank the reviewer for his/her appreciation of our work

This is shown by immunofluorescence staining as well as proteomic analysis of the biotinylated proteins. The images in figure 4b (and to lesser extent 4c) show an increased intensity and more diffuse labelling. Can the authors exclude that RNase treatment simply leads to an increase in accessibility for the biotin-phenol, hence a visibly higher biotinylation?Along these lines, have the authors maybe observed an increase in overall labelling/pulldown efficiency or for biotinylated proteins in their proteomic data?

We appreciate the comment of the reviewer. While we cannot fully exclude the possibility that RNase treatment results in an increased accessibility for biotin-phenol, we do not think that this is the case. We neither observe a general increase in pulldown efficiency of biotinylated proteins nor do we see a substantially increased biotin staining per nucleus. In addition, we actually see more proteins being less efficiently labelled upon RNase treatment suggesting that the domain is actually more dispersed in the nucleus.

Minor comments:1. In figures 1a and 4a (as well as in the Methods section), the authors use the term 'biotin-tyramide' as labelling agent, but in the main text and figure legends 'biotin-phenol' is used. For clarity, only one term should be used.

We use Biotin-phenol throughout the manuscript when labelling proteins and biotin-anilin to label RNA

2. Figure 2a shows a magnified cell/nucleus in the last column. To what cells do the magnifications in this last column refer to? Maybe these cells could be boxed in the second last column?We thank the reviewer for this suggestion and boxed the cells that are magnified in figures 1 and 2 as well as the new figure 53. In figures 4b + c:, the figure legend mentions the individual rows as '(I)' and '(II)' but no such label seen in the corresponding panel(s).

We apologize for the mishap and added the corresponding row labels

4. The Quantification method for co-localization (e.g. 1c and 2b) is insufficiently described to the reader (reference simply relates to Fiji package). What module/script within the Fiji package has been used?

We now described the version of the Fiji package we used for colocalization analysis in the methods section

5. The RNase treatment is not described at all in the methods section or the supplementary information and should be added.

We apologize for the mishap and added the Rnase treatment in the method section.

6. The sentence on page 6 ('As expected, neither the targeted signature factor or proteins that mainly interact with them protein-protein interactions such as MSL1,3 and MOF for MSL2 or Cenp-C for Cid are not affected by RNAase treatment') should be rephrased as it is not comprehensible in the current form.

We have changed the sentence to: “As expected, neither the targeted signature factor or proteins that mainly interact with them by protein-protein interactions such as MSL1,3 and MOF for MSL2 or Cenp-C for Cid are affected by RNAase treatment.”

Reviewer #2:This manuscript by Choudhury et al. describes a new method for antibody-mediated proximity labeling and applies it in the cell nucleus. In short, nuclei are isolated, fix and permeabilized, proteins are labeled with primary antibodies, a bacterially expressed/purified protein-A-APEX2 fusion protein is added, conventional H_2_O_2_/biotin phenol labeling of proximate protein is performed, proteins are un-crosslinked and biotin-affinity captured for MS analysis. The application to nuclear proteins and results seems appropriate. The method is highly similar to and more complicated than prior methods as described in more detail below. I would focus the impact of this paper towards its biological results and not the novelty of the methods used.Prior methods that effectively accomplish the same outcome (fixed cells/tissues, antibodies and proximity labeling for AP-MS) have been published before. Perhaps most recently it was reinvented as the so called BAR method in PMID 29256494. That paper was cited here but incorrectly as BirA-related, which it is not. Of course that prior manuscript itself ignored prior methods from years back (2008, 2012, 2014, 2015, PMID 18495923, 22936677, 24706754, 25829300) using the same approaches of antibody targeted peroxidase for the same purposes of proximity labeling.This method seems a somewhat Rube Golderbergian approach to antibody-mediated proximity labeling, which has been performed previously in multiple reports. APEX/2was developed to function inside of living cells since HRP does not. The value of doing the proximity labeling in living cells was either to capture protein associations over time, as with BioID/TurboID, or to get snapshots of protein associations in living cells with APEX/2. HRP does however function quite well for proximity labeling outside of cells, or in fixed/permeabilized cells, as has been demonstrated in the prior methods/papers that are referenced above. Replacing commercially available secondary antibodies fused to HRP with homemade protein-A-fused to APEX2 seems counterintuitive and/or unnecessary.

We agree with all four reviewers that while we optimized the method and showed its high sensitivity and versatility similar methods have been published before. We have now discussed the various approaches more extensively in the revised manuscript. However, we would also like to point out that our method allows the purification of proteins and RNA in proximity to specific bait proteins using the same highly sensitive tools we describe. Moreover, the application of these method now revealed a more detailed characterization of the centromeric chromatin in *Drosophila* and allowed us to describe for the first time the DHX9 dependent R-loop formation at the *Drosophila* centromere. Following the suggestion of reviewer 2, we now put a stronger focus on the biological results rather than the method used.

Could the authors explain the mechanisms that underly the reported enhanced sensitivity of AMPL-MS compared to conventional APEX2 in living cells. Is there something about the nuclear isolation that reduces interfering background, the loss of small soluble molecules in the nucleus after isolation and/or permeabilization that enhance the proximity labeling, penetration issues with the biotin-phenol in living cells, and/or something else?

We have now added a paragraph describing why we think the method is more sensitive. Indeed, we think most of the aspects suggested by the reviewer contribute to it. However, based on our experience we think the strongest contribution is the fact that we isolate and permeabilize the nuclei, which greatly enhance the accessibility of biotin-phenol.

There seems to be the use of various controls based on the figures and legends, but they are not clearly described in the results or methods.

We have extensively described the controls used in the manuscript to better illustrate the rationale behind them

All MS results should be provided, preferably in an Excel file format.

All MS results are provided in an.xls format (Supplementary files S1-S11)

applied.Reviewer #3The manuscript entitled 'The role of RNA in the maintenance of chromatin domains as revealed by antibody mediated proximity labelling coupled to mass spectrometry' by Choudhury et al. describe a new method, which they termed AMPL-MS (Antibody mediated proximity labelling mass spectrometry). The technique is based on proximity labelling but uses antibodies instead of fusion proteins. They use this method to characterize chromatin domains containing specific signature proteins or histone modifications and focus on the composition of chromocenter as well as the chromosome territory containing the hyperactive X-chromosome in *Drosophila*. Last but not least they include data that show that RNA is involved in maintaining the integrity of chromatin domains by RNAse treatment and mass spec analysis.The technique works well and the results are very clear. I therefore expect that, in the right hands, it is very reproducible.

We thank the reviewer for his/her appreciation of our work

There are a few points that the authors may want to address:1. Title'The' role of RNA in the maintenance of chromatin domains as…, seems too much of a statement. The title is therefore an overstatement that needs to be fixed.

While we agree with the reviewer that the title was a bit of an overstatement in the initial version of the manuscript we now feel that it very much reflects the additional findings we present in the revised manuscript.

2. Figure 1In Figure 1 the authors show very convincingly that the methods works well in their hands. They report on 172 proteins that localized in proximity to CID containing centromeric chromatin but do not provide the list of proteins as far as I can tell. Especially the RNA binders should be named.

We have extended our discussion of the proteins in proximity to CID focussing on the proteins known to bind RNA. This is especially interesting as many of them have been shown or were at least suggested to play a role in the formation or resolution of R-loops.

3. Figure 2Using the hyperactive X is very clever when addressing RNA function but it should be stated in the discussion that there may be certain aspects that are specific to the male x and that is impossible to discriminate general and specific effects uncovered by this method.

We have added a sentence pointing out that we only can not distinguish between a specific and MSL2/rox dependent effect and a general effected caused by the upregulation of transcription on the male X-chromosome.

4. Figure 3The authors should state more clearly the new findings of this figure since it is not fully obvious from its current representation.

Such a detailed analysis of differentially modified chromatin domains has not yet been done in the *Drosophila* system and therefore provides a rich resource for further analysis. Moreover, the highly sensitive AMPL-MS method also allowed us to investigate several novel factors involved in splicing and RNA processing that are selectively detected in the neighbourhood of H3K9me3 or H4K16ac containing chromatin, pointing towards a major and general role of RNA in the organisation of chromatin.

5. Figure 4These are certainly interesting data but the authors remain in the very descriptive state. This is fine for a methods paper but then, the authors should hypothesize more on what the results mean. Are certain RNA dependent factors specific or general and they then recruit a specific set of factors that fall off upon RNAse treatment as a secondary effect or because they bind RNA directly. I feel like there may be more information that they authors got get out of there data than what they currently provide.

We agree with the reviewer that the data carry much more information that we can describe or analyse in a single paper. To dig a bit deeper into the data set we started to characterize the RNA that is specifically enriched in proximity to a particular chromatin domain and investigate a novel and yet unknown centromeric function of the RNA helicase MLE/DHX9 in *Drosophila*. In fact, we think our finding provides a possible function for the widely observed centromeric transcription. We purpose that stable R-loops might help recruiting helicases such as MLE to the centromere to prevent an excessive formation of a non canonical DNA structure that interferes with the assembly of centromeric nucleosomes.

6. DiscussionThe authors state: 'While we have not identified the RNAs responsible for the formation of theses domains, we clearly observe that they do confer specificity for the domains as we observe very little overlap in the factors lost from the corresponding domains (Figure 4h). the 'specificity' is hard to determine since factors bound to these regions are different, and therefore different factors will fall off, regardless of whether the RBP are specific unless the RNA is involved in recruiting the factors specifically, which the authors have not shown. Therefore, this result is suggestive and interesting but the statement is too strong and not backed by their results.

In the revised version of the manuscript we now show that the RNAs composition is different among the two different domains studied. We agree with the reviewer that the data carry much more information that we can describe or analyse in a single paper. To dig a bit deeper into the data set we started to characterize the RNA that is specifically enriched in proximity to a particular chromatin domain and investigate a novel and yet unknown centromeric function of the RNA helicase Mle/DHX9 in *Drosophila*. In fact, we think our finding provides a possible function for the widely observed centromeric transcription. We purpose that stable R-loops might help recruiting helicases such as MLE to the centromere to prevent an excessive formation of a non canonical DNA structure that interferes with the assembly of centromeric nucleosomes.

Overall, this is an interesting method that has been used in the past to identify protein modifications with high quality antibodies available. The authors show here that the method can also be used to different nuclear proteins and detect changes in protein complex composition. As it is it is primarily a methods paper, and for that the results are very clear. Gain of new info is not large but it is a useful technique to continue research on this subject and is a nice start of many new avenues into how RNA effects chromatin.

We would like to thank the reviewer for his/her positive remarks

Reviewer #4This is a short report featuring an innovative proximity labeling approach to the identification of proteins enriched in distinct types of chromatin domains. The domains compared are centromeric heterochromatin and X-linked hyperactive chromatin in *Drosophila* cells. These are relatively well-described domains, thus serving as an excellent test for the targeting of biotinylation in the permeabilized nucleus via interaction of specific antibodies with ProteinA-Apex2 provided exogenously. In parallel with the signature chromatin proteins CID or MSL2 as baits, the authors also target proteins in proximity to specific histone tail PTMs. Taking the work one step further, they compare the recovery of proteins +/- pretreatment of nuclei with RNase. They conclude that in each case selective interactions are specifically lost with pre-treatment of RNase.Major comment:As mentioned above, the approach is innovative and raises the possibility of a simpler MS method to identify protein-protein interactions. The RNase result is also provocative. However, in each case the specificity of potentially novel results are not explored further. Thus, the work is of interest but clearly still preliminary.Did the authors dig deeper into novel interactions without obtaining convincing validation? Did they conclude that the MS approach is worth pursuing further or not? Admittedly the RNase result is difficult to follow up, but additional discussion of prior related work as well as consideration of future experiments would help improve the manuscript.

Following the reviewers suggestion we have now further extended our investigation towards the RNA in proximity to the corresponding bait proteins. This allowed us to show that the RNAs in proximity to the baits are indeed transcribed from the DNA that constitutes the chromosomal domains. Together with the proteins found in proximity we hypothesized that the RNAs are involved in the formation of R-loops around the centromer, which we could validate using a new probe for R-loops in *Drosophila*.